# Recurrence intervals for the closure of the Dutch Maeslant surge barrier

Henk W. van den Brink[1] and Sacha de Goederen[2]

[1]KNMI, Utrechtseweg 297, De Bilt, The Netherlands.
[2]Rijkswaterstaat, Boompjes 200, Rotterdam, The Netherlands.

*Correspondence to:* henk.van.den.brink@knmi.nl

**Abstract.** The Dutch Maeslant-barrier, a movable surge barrier in the mouth of river Rhine, closes when due to a surge in the North Sea the water level in the river in Rotterdam would exceed 3 m above mean sea level. An important aspect of the failure probability is that the barrier might get damaged during a closure and that, within the time needed for repair, as a second critical storm-surge occurs. With an estimated closure frequency of once in 10 years, the question arises how often the barrier has to

be closed twice within a month.

Instead of tackling this problem by the application of statistical models on the (short) observational series, we solve the problem by combining the surge model WAQUA/DCSMv5 with the output of all seasonal forecasts of the European Centre of Medium-Range Weather Forecasting (ECMWF) in the period 1981-2015, whose combination cumulates in a pseudo-observational series of more than 6000 years.

We show that the Poisson process model leads to wrong results, as it neglects the temporal correlations that are present on daily, weekly and monthly scales.

By counting the number of double events over a threshold of 2.5 m, and using that the number of events is exponentially related to the threshold, it is found that two closures occur on average once in 150 years within a month, and once in 330 years within a week. The large uncertainty in these recurrence intervals of more than a factor of two are caused by the sensitivity of

the results to the Gumbel parameters of the observed record, which are used for bias correction.

Sea level rise has a significant impact on the recurrence time, both of single and double closures. The recurrence time of single closures doubles with every 18 cm mean sea level rise (assuming that other influences remain unchanged), and double closure duplicate with every 10 cm rise. This implies a 3-14 times larger probability of a double closure for a 15-40 cm sea level rise in 2050 (according to the KNMI climate scenarios) than that is currently dealt with.

**Keywords.** Maeslant-barrier

    surge

    tide

    sea level rise

    failure probability

ECMWF

    seasonal forecasts

# 1 Introduction

In 1953, a large part of south-west Netherlands was flooded by the sea, with over 1800 casualties. After these floodings it was decided to shorten the Dutch coastline by approximately 700 km by building both closed and permeable dams between the isles in the south-west of the country. In this way not all dikes had to be made higher.

In 1987 it was decided to build a movable surge barrier in the so-called New Waterway (in Dutch: *'Nieuwe Waterweg'*, which is the artificial mouth of the river Rhine into the North Sea, located at 20 km downstream from the city of Rotterdam) which has only to be closed during dangerous situations. In this way, the Rotterdam harbour can remain accessible for sea shipping. This barrier, called the Maeslant-barrier (in Dutch: *'Maeslantkering'*), has been operational since 1997. When the forecasted water level in Rotterdam exceeds 3 m above NAP (Normaal Amsterdam Peil: 'Amsterdam Ordnance Datum', which is approximately equal to mean sea level), the barrier is closed[1]. This situation was expected to happen once in approximately 10 years. In the period 1997-2016, the barrier has been closed once in storm conditions: this event happened on 8-9 November 2007.

In order to guarantee the required safety level for the hinterland, the failure probability of the Maeslant-barrier is required to be maximally 0.01, i.e., it has to close correctly in 99 of the 100 cases (Rijkswaterstaat, 2013). An important aspect of the failure probability is the scenario that the barrier gets damaged during a closure and that, within the time needed for repair, a second critical storm-surge occurs.

The time that the barrier can not be closed due to repair depends, naturally, on the complexity of the breakdown. Therefore we explore the frequency of all succeeding closures with an inter-arrival time from one day to one month.

For the estimation of the probability of two closures within a given short time interval (which we here will call a double closure), the observational record of the one single event that did occur obviously does not provide any information about inter-arrival times. In order to derive nevertheless information about the double closures from the observations, one possibility could be to explore how often a threshold lower than 3 m above NAP has been exceeded, and to scale these probabilities to the required level. A different approach might be to regard the closures to be independent, which leads to a Poisson distribution for the inter-arrival times (see Section 3). Using that the average return period is about 10 years, an estimate can be obtained how often the recurrence time is one week or a month. However, the result of this approach is very sensitive to the estimated recurrence period, and is biased due to the neglection of temporal correlations in the atmosphere.

We therefore used an alternative approach (Van den Brink et al., 2005b), i.e., by combining the seasonal forecasts (Vialard et al., 2005) of the European Centre of Medium-Range Weather Forecasting (ECMWF) into a large dataset, representing the current climate with more than 6000 independent years (up till December 2015). Thereafter we calculated the surges from the winds and pressures from this dataset, resulting in a high-frequent time series of water levels with the same length as the ECMWF dataset. From this dataset of water levels, the required inter-arrival times in a stationary climate can be counted and analysed.

---

[1]Formally, the barrier also closes if the level in Dordrecht exceeds 2.90 m above NAP. However, it is very unlikely that the water level exceeds the threshold in Dordrecht but not in Rotterdam.

The paper is structured as follows: the meteorological and hydrological models, as well as the observational dataset, are described in Section 2. Section 3 explains the applied methodology, Section 4 shows the validation of the model outcomes, and Section 5 describes the results. The conclusions are presented in Section 6.

## 2    Models and observations

The seasonal forecasts of the ECMWF are used to drive the surge model WAQUA/DCSMv5, which outputs (among others) the water level at the coastal station Hoek van Holland (see Figure 4 for its location). The city of Rotterdam is located about 25 km from Hoek van Holland upstream of the river Rhine. Although the height of the water level in Rotterdam is mainly determined by the water level at Hoek van Holland, it is also influenced by the discharge of the river Rhine. A simple analytical relation is therefore used to simulate the effect of the Rhine discharge on the water levels in Rotterdam. All three models are briefly described below. Also the observational record is described.

### 2.1    ECMWF seasonal model runs

From November 2011 onward the ECMWF produces every month an ensemble of 51 global seasonal forecasts up to 7 months ahead, i.e. amply surpassing the 2 weeks horizon of weather predictability from the atmospheric initial state. Over the period 1981–2011, re-forecasts with smaller ensembles have been performed to calibrate the system. The forecast system consists of a coupled atmosphere–ocean model. The atmospheric component has a horizontal resolution of T255 (80 km) and 91 levels in the vertical (Molteni et al., 2011). The ocean component NEMO has a resolution of 1 degree and 29 vertical levels (Madec, 2008). The wave model WAM (Janssen, 2004) allows for the two-way interaction of wind and waves with the atmospheric model. All forecasts are generated by the so-called System 4 (Molteni et al., 2011).

The ECMWF dataset provides, among other fields, global fields of 6-hourly wind and sea-level pressures (SLP). We have regridded the data to a regular grid of 0.5 degrees.

From every 7-month forecast, we skipped the first month in order to remove dependence between the perturbed members due to the correlation in the initial meteorological states. Van den Brink et al. (2005a) show that the correlation of the NAO index approaches zero for the forecasts after 1 month. We combined two forecasts that differ 6 month in start time to construct a full calendar year. The total number of forecasts that have been combined to full years is 12556, resulting in 6282 independent calendar years. Table 1 clarifies how the individual members are combined to construct the 6282-year timeseries. It shows that the first year is constructed from the combination of ensemble member 0 starting in January 1981 with ensemble member 0 starting in July 1981. As the first months are skipped, together they cover the period 1 February 1981 to 31 January 1982. The next year continues with 1 February 1982.

Although the thus obtained dataset is as continuous as possible, several peculiarities are left. First, there is a discontinuity at every concatenation point, which aborts the temporal correlation in the meteorological situation. The correlation in the astronomical tide is however preserved. As the concatenation follows the historical order for every perturbation number, possible low-frequent variability (e.g. due to the sea surface temperature) is maintained (Graff and LaCasce, 2012). In this way, the

**Table 1.** Combination of individual forecast members to construct the 6282-year time series. The numbers indicate the startyear (1981-2015), followed by the perturbation number (0-50). See the text for more explanation.

| year | first half year | + | second half year |
|---:|---|:-:|---|
| 1 | jan 1981-0 | + | jul 1981-0 |
| 2 | jan 1982-0 | + | jul 1982-0 |
| ⋮ | | | |
| 35 | jan 2015-0 | + | jul 2015-0 |
| 36 | jan 1981-1 | + | jul 1981-1 |
| ⋮ | | | |
| 669 | jan 2015-50 | + | jul 2015-50 |
| 670 | feb 1981-0 | + | aug 1981-0 |
| ⋮ | | | |
| 6282 | jun 2015-50 | + | dec 2015-50 |

18.6-year lunar nodal cycle is also incorporated. The only discontinuities in the initial states occur when 2015 is reached and the next year starts again in 1981 (from year 35 to 36 in Table 1). Discontinuities in the calendar years are made when the perturbation number jumps back from 50 to 0. In that case, one calendar month is skipped (from year 669 to 670 in Table 1).

5 These few discontinuities have negligible influence on the outcomes.

## 2.2 WAQUA/DCSMv5 model

To infer surge heights in the North Sea from the ECMWF output we use WAQUA/DCSMv5 (Gerritsen et al., 1995). This model solves the two-dimensional shallow-water equations on a $\frac{1}{12} \times \frac{1}{8}$ (approximately $8\,\text{km} \times 8\,\text{km}$) grid on the Northwest European shelf region. It is operationally used at KNMI to predict the water levels along the Dutch coast. Meteorological input are SLP

10 and 10-m wind. The latter is translated into wind stress using a drag coefficient based on the parametrization of Charnock (1955), with a Charnock parameter of 0.032. The astronomical tide is prescribed at the open boundaries in ten harmonic constituents ($O_1, K_1, N_2, M_2, S_2, K_2, Q_1, P_1, \nu_2$ and $L_2$) and propagates from there into the model domain. The model output consists of total water level and the height of the astronomical tide in the absence of meteorological forcing.

We analyse the model results in terms of total water level as this is the quantity relevant for the closure of the Maeslant

15 barrier.

## 2.3 Rhine discharge model

The water level at Rotterdam is influenced both by the sea level at Hoek van Holland and the river discharge. Based on calculations by Rijkswaterstaat (De Goederen, 2013, page 25), the water level at Rotterdam $L_R$ can be approximated by:

$$L_R - L_{HvH} = 4.08 \cdot 10^{-5}(Q_L - 1750) \tag{1}$$

in which $L_{HvH}$ is the level at Hoek van Holland, and $Q_L$ is the river Rhine discharge at Lobith (where the Rhine enters The Netherlands, see Figure 4) in $\mathrm{m}^3\mathrm{s}^{-1}$.

In order to take the effect of the river discharge into account, we applied the right hand side of Eq 1 to the historical 1901-2000 daily record of discharges at Lobith. To every WAQUA/DCSMv5 ensemble member timeseries for Hoek van Holland we added a 6-month period starting with the same date as the ECMWF ensemble member starts with, randomly selected from the Lobith record. In this way the seasonal variation of the river discharge is maintained. This approach implies that there is no correlation between high sea surges and river discharges, which is approximately true (Van den Brink et al., 2005a; Kew et al., 2013)[2].

## 2.4 Water level observations

The observational record of water levels at Hoek van Holland starts in 1864 (Holgate et al., 2013; PSMSL, 2017). Accurate readings of the water level start in August 1887. We used the data from 1888 onward. The data before 1987 are obtained visually from (digitized) charts, afterwards 10-min average values are used.

Due to sea level rise and land subsidence, the observational water levels in a historical record have to be corrected for these influences. Figure 1 shows how the observations taken in Hoek van Holland from 1880 to 2015 have to be adjusted to be representative for the year 2009. The correction varies between for 31 cm for 1888 to -2 cm for 2015. The rapid change around 1965 and the change in the slope can be attributed to the extension of the Rotterdam harbour to the west (Dillingh et al., 1993; Hollebrandse, 2005; Becker et al., 2009). The annual maxima of the water level in Hoek van Holland are shown in Figure 2, both uncorrected and corrected. The observational record contains (after correction) 10 events that exceed 3 m in Hoek van Holland, the smallest inter-arrival time being 1.2 years (in 1953 and 1954). This makes direct derivation of the recurrence intervals of inter-arrival times smaller than 1 year impossible.

## 3 Methodology

In this section the methodology is presented that we use to derive the recurrence times of double closures within one month. As mentioned earlier, the observational record does not contain double closures within one year, which makes direct derivation of the required inter-arrival times impossible. We therefore combine the observations with information from the ECMWF-WAQUA/DCSMv5 dataset. The first step in the evaluation of the quality of this dataset is to check whether the annual maxima of the water level in Hoek van Holland as derived from the ECMWF and WAQUA/DCSMv5 dataset has the same distribution the observational dataset. The theory of the annual maxima, which is applied for the intercomparison, is described in section 3.1.

The next step is to check whether the recurrence times of double closures can be described by a Poisson distribution, which would occur if the events are mutually independent (section 3.2). Section 5 shows that the events are not mutually independent,

---

[2]In the case that extreme discharge and extreme water levels are correlated, the most promising solution - in line with the topic of this paper - is to use the precipitation amounts, the temperature and snow melt in the Rhine basin as input for a hydrological model to calculate the Rhine discharge. In this way no explicit assumptions about the correlation have to be made. This is however outside the scope of this paper.

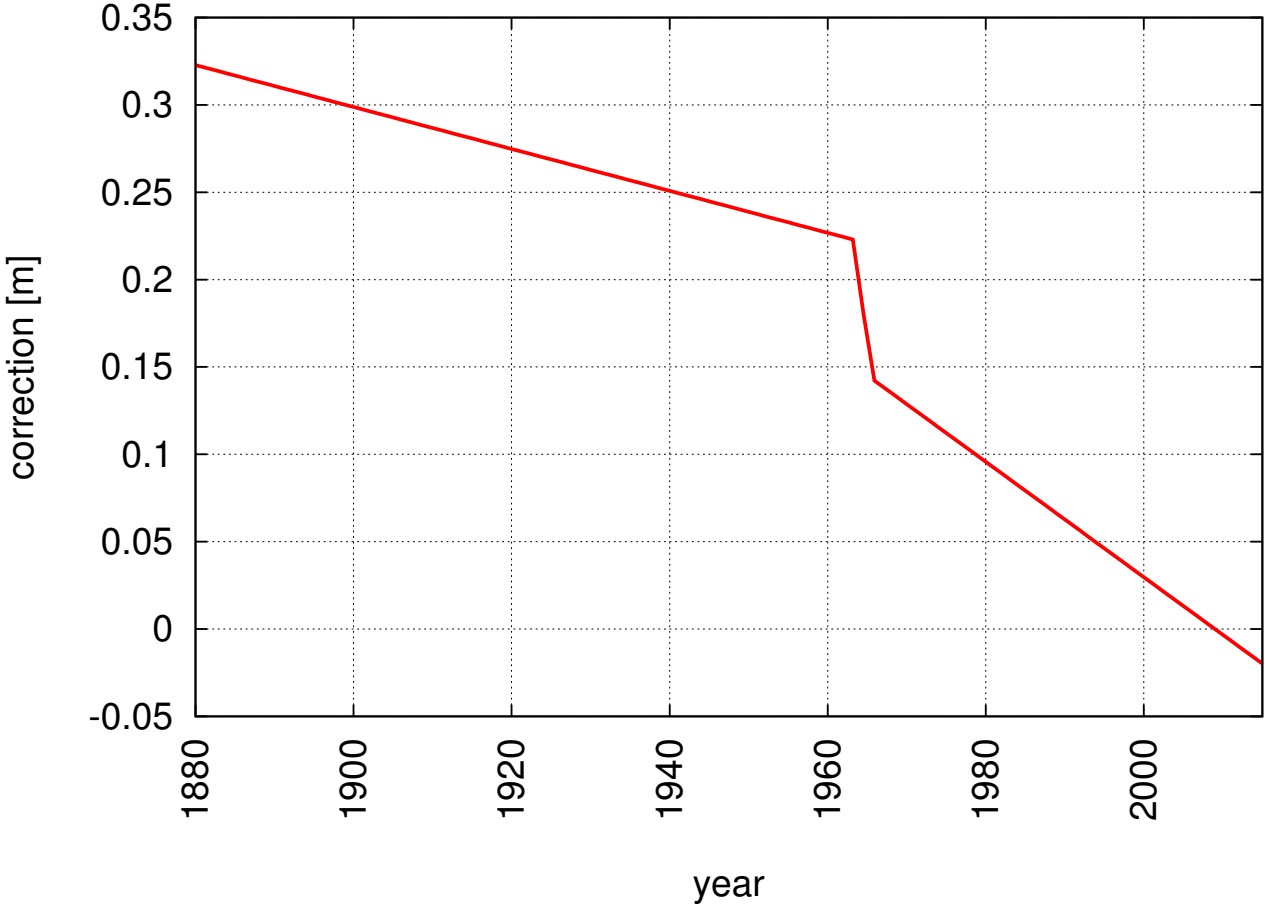

**Figure 1.** Adjustment of the observed sea levels in Hoek van Holland (1880-2015) to correct for sea level rise and land subsidence. The observational record is adjusted to be representative for the situation in 2009.

which hinders the application of the Poisson distribution, either to the observations or to the ECMWF-WAQUA/DCSMv5 dataset.

The huge size of the ECMWF-WAQUA/DCSMv5 dataset invites us to explore whether the recurrence times of the events

5   that exceeds the threshold of 3.0 m can be derived directly from the empirical density function (EDF) of the dataset. This EDF is introduced in section 3.3. Section 5 shows, however, that even the 6282-year dataset is not long enough for an accurate estimation of the desired recurrence times by counting the intervals. We introduce an extra step by using a lower threshold than 3.0 m, and by deriving the relation between the threshold and the number of inter-arrival times (Eq 11).

Section 4.2 shows that a small bias correction of the ECMWF-WAQUA/DCSMv5 dataset is necessary (Eq 10). A short

10   analysis of the uncertainty analysis introduced by this bias correction, as well as by the use of a lower threshold, are given in Section 3.4.

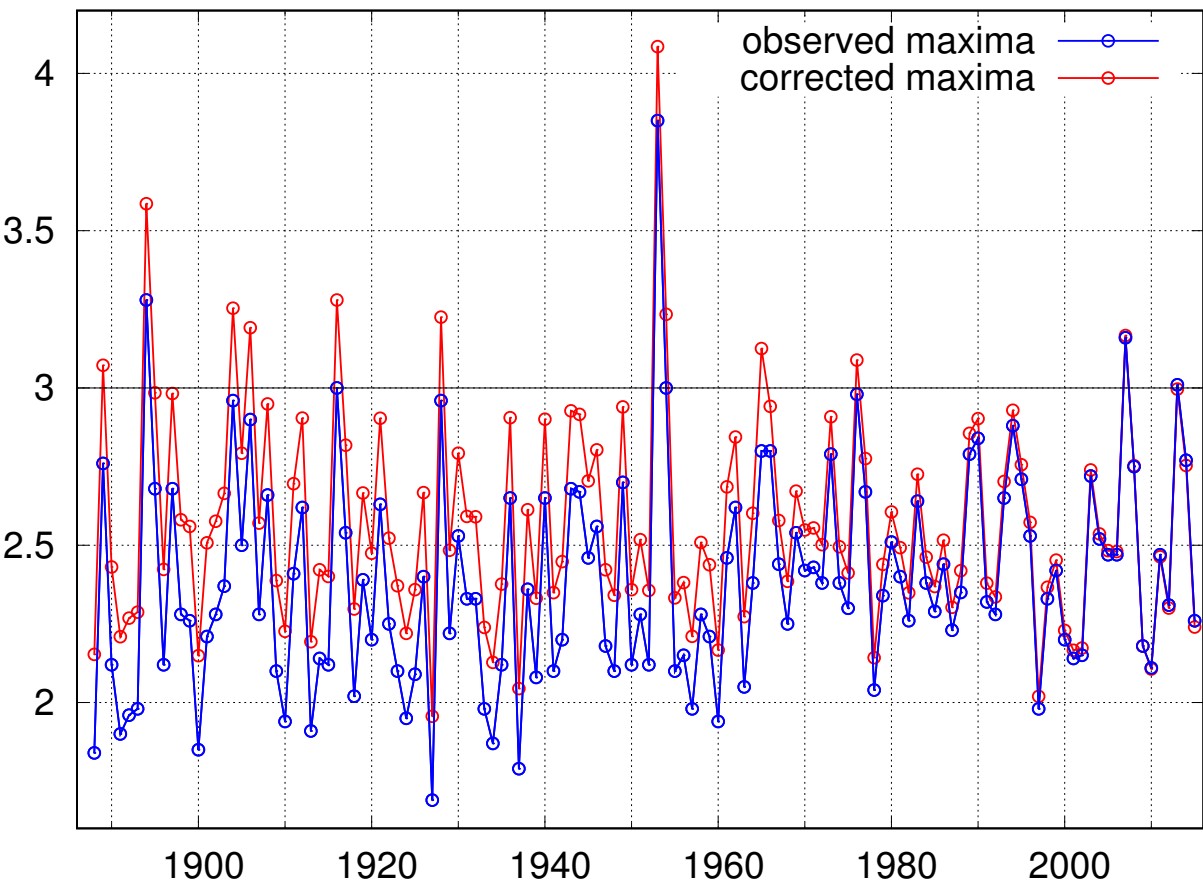

**Figure 2.** Annual maxima of the observed water levels in Hoek van Holland, 1887-2015 (blue). Correction according to Figure 1 leads to the red line.

### 3.1 Extreme value analysis

To determine the extreme water levels that occur on average once in a given period (the return period), annual maxima are fitted to a Generalized Extreme Value (GEV) distribution, which is the theoretical distribution for block maxima (e.g. Coles, 2001):

$$G(y) = \exp\{-[1 + \xi(\frac{y - \mu}{\sigma})]^{-1/\xi}\} \tag{2}$$

Here $\mu$, $\sigma$ and $\xi$ are called the location, scale and shape parameter, respectively, and $y$ is the sea water height . If $|\xi| \to 0$, Equation 2 can be written as

$$G(y) = \exp\{-\exp[-\frac{y - \mu}{\sigma}]\} \tag{3}$$

which is called the Gumbel distribution.

The return period $T_s$, which is the average recurrence time of a single exceedence of level $y$, is defined by

$$T_s = \frac{1}{1 - G(y)} \tag{4}$$

For large return periods, the combination of Eq (3) and (4) can be approximated by:

$$y \approx \mu + \sigma \log(T_s) \tag{5}$$

The distributions of the annual extremes are in this paper presented in the form of Gumbel plots, in which the annual maxima (or minima) are plotted as a function of the Gumbel variate $x = -\ln(-\ln(G(y)))$. In case of a Gumbel distribution this results in a straight line. Via Eq (4) the Gumbel variate is directly related to the return period, which we label on the upper horizontal axis of the plots.

The parameters are derived by maximum likelihood estimation.

## 3.2   Interarrival times

The inter-arrival times of independent events can be described as a Poisson process. If $N_t$ is the number of events that occurs before time $t$, and $1/\lambda$ is the average recurrence time, then

$$P(N_t = k) = \frac{(\lambda t)^k}{k!} e^{-\lambda t} \tag{6}$$

We are interested in the probability that the time until the next event $\Delta T$ is larger than a given value $t$. This means that no events occurred before time $t$, i.e. $k = 0$. It thus follows that

$$P(\Delta T > t) = P(N_t = 0) = e^{-\lambda t} \tag{7}$$

which states that the inter-arrival time between independent events is exponentially distributed.

For small inter-arrival times ($\Delta T \ll 1/\lambda$), Eq (7) can be rewritten as:

$$T_d \approx \frac{1}{\lambda \Delta T} \tag{8}$$

in which $T_d$ is the recurrence time of a double event. An average recurrence time of 10 years thus implies that a double closure within a month occurs once in 120 years if independence is assumed.

## 3.3   Empirical Distribution Function

Let $x_1 \leq x_2 \ldots \leq x_n$ be $n$ observations from distribution $F$, then the Empirical Distribution Function (EDF) $\hat{F}$ is given by (e.g. Buishand and Velds, 1980):

$$\hat{F}(x_i) = \frac{i}{n+1} \tag{9}$$

Eq (9) states that $F(x)$ can be estimated from the number of observations lower than $x$. The advantage of Eq 9 is that it requires no assumptions about $F$.

## 3.4 Uncertainty analysis

We consider two contributions to the uncertainty in the estimation of the recurrence intervals. The first one is the bias correction in the location- and scale parameter of the GEV distribution (Eq 10). As the ECMWF-WAQUA/DCSMv5 dataset is 49 times longer than the observational record, the uncertainty in the bias correction will be dominated by the uncertainty in the Gumbel fit to the observations. A first-order estimation of the 95% uncertainty range due to the bias correction is made by keeping the ECMWF-WAQUA/DCSMv5 unchanged in Equation 10, and replacing $\mu_{obs}$ and $\sigma_{obs}$ with $\mu_{obs} \pm 2\Delta\mu_{obs}$ $\sigma_{obs} \pm 2\Delta\sigma_{obs}$, respectively. Here, $\Delta\mu_{obs}$ and $\Delta\sigma_{obs}$ are the standard errors in the location- and scale parameter as derived from the observations. Redoing the calculations with those adjusted bias corrections gives a good indication of the uncertainty range in the estimated recurrence intervals.

The second contribution to the uncertainty in the estimation of the recurrence intervals is the choice of the threshold and its scaling to the threshold of 3.0 m (Eq 11). The range of thresholds is varied from 2.3 m (high enough to resemble extreme conditions) and 2.7 m (low enough to reduce statistical uncertainty). The variation of the recurrence intervals for different thresholds gives an indication of the sensitivity of the estimated recurrence intervals for the choice of the threshold.

# 4 Evaluation

## 4.1 ECMWF wind and pressure fields

In order to model extreme surge events correctly, in particular the wind and pressure should be well represented by the model. Due to the sensitivity of model wind to the drag parametrisation, it is difficult to verify the model winds directly. Instead, we validated the SLP. This direct model parameter can be compared more easily with observations than wind data, and is a good measure of the capability of the model to produce deep depressions (see also Sterl et al., 2009).

Figure 3 shows the annual minimum daily-mean SLP for the observations in Nordby, Denmark (8.4°N,55.45°N) for the 1874-1986 period, and the ECMWF data for the nearest grid point. This location was chosen because a pressure minimum in this area leads to long North-West oriented wind fetches over the North Sea and therefore to high surges at the Dutch coast. This is illustrated in Figure 4, which depicts the pressure and the wind field related to the highest surge of 4.29 m in Hoek van Holland that occurred on 21 January 1988, in ensemble member 17 that started on 1 August 1987. The figure also depicts the location of Nordby.

In the Gumbel plot (Figure 3) observed and simulated values yield parallel curves. Modelled pressures are slightly lower than the observed ones, but the model has the same relation between intensity and frequency of low pressures as the observations have. There is no sign of an artificial lower limit on pressure in the model. From Figure 3 we conclude that the ECMWF dataset is appropriate to drive a surge model for water level calculations.

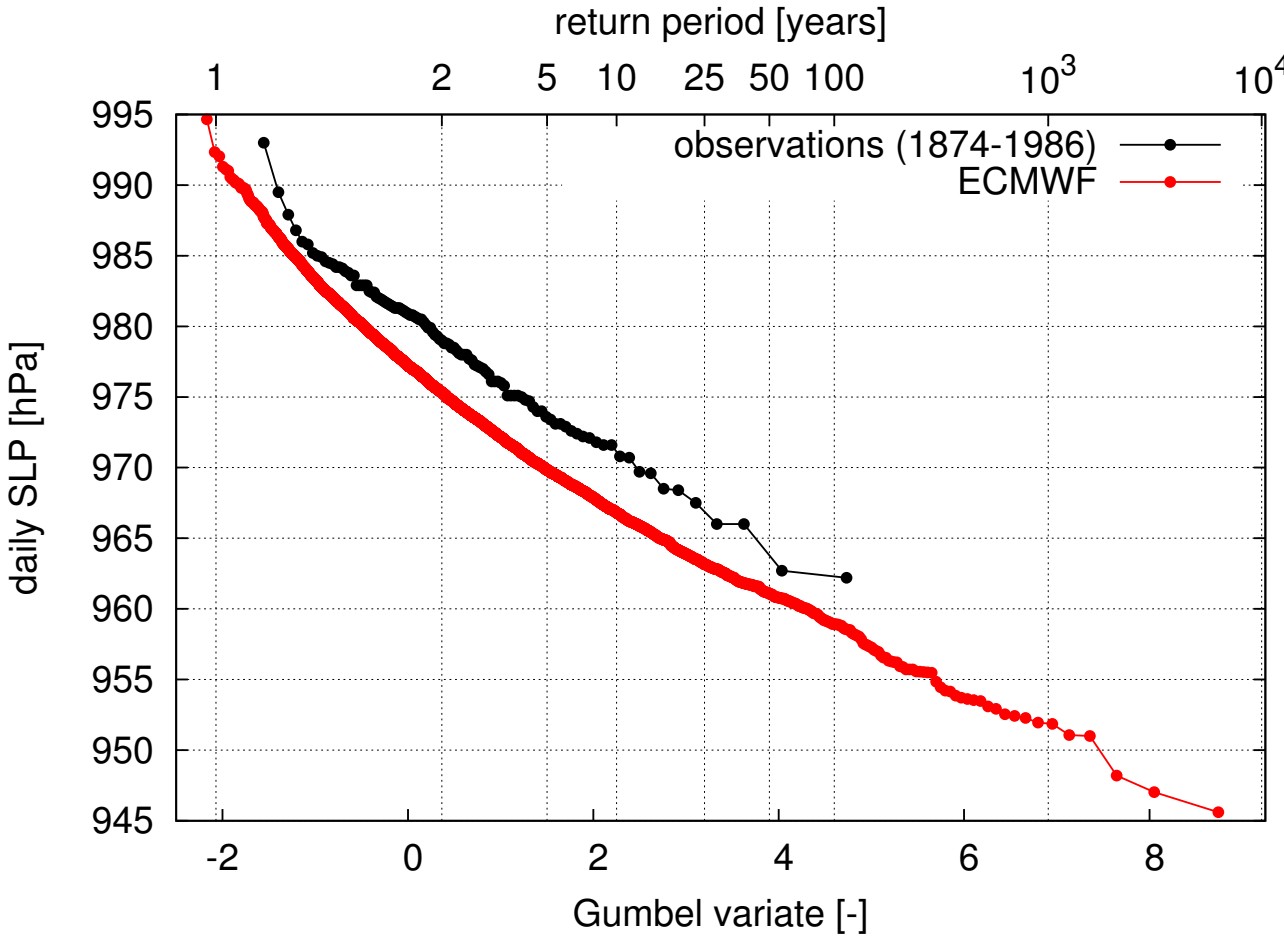

**Figure 3.** Gumbel plot of observed annual-minimum sea level pressure in Nordby (black) and as simulated at the nearest ECMWF grid point (red). The location of Nordby is indicated in Figure 4.

### 4.2   WAQUA/DCSMv5

Figure 5 shows the Gumbel plot of the annual maxima for the observations (black) and the ECMWF-WAQUA/DCSMv5 ensemble (green) for Hoek van Holland. The once-a-year extreme water level of 2.43±0.02 m (represented by the Gumbel location 5   parameter) is reproduced within 0.6% (2.36±0.004). The scale parameter of the Gumbel distribution (0.264±0.018) is slightly underestimated (0.252±0.002). It is likely that this underestimation is caused by the fact that WAQUA/DCSMv5 uses a fixed Charnock parameter, whereas the ECMWF uses a time-varying Charnock parameter. For high winds the ECMWF Charnock parameter exceeds the value of 0.032 used by WAQUA/DCSMv5, which leads to underestimation of high surges (Zweers et al., 2010; Van Nieuwkoop et al., 2015).

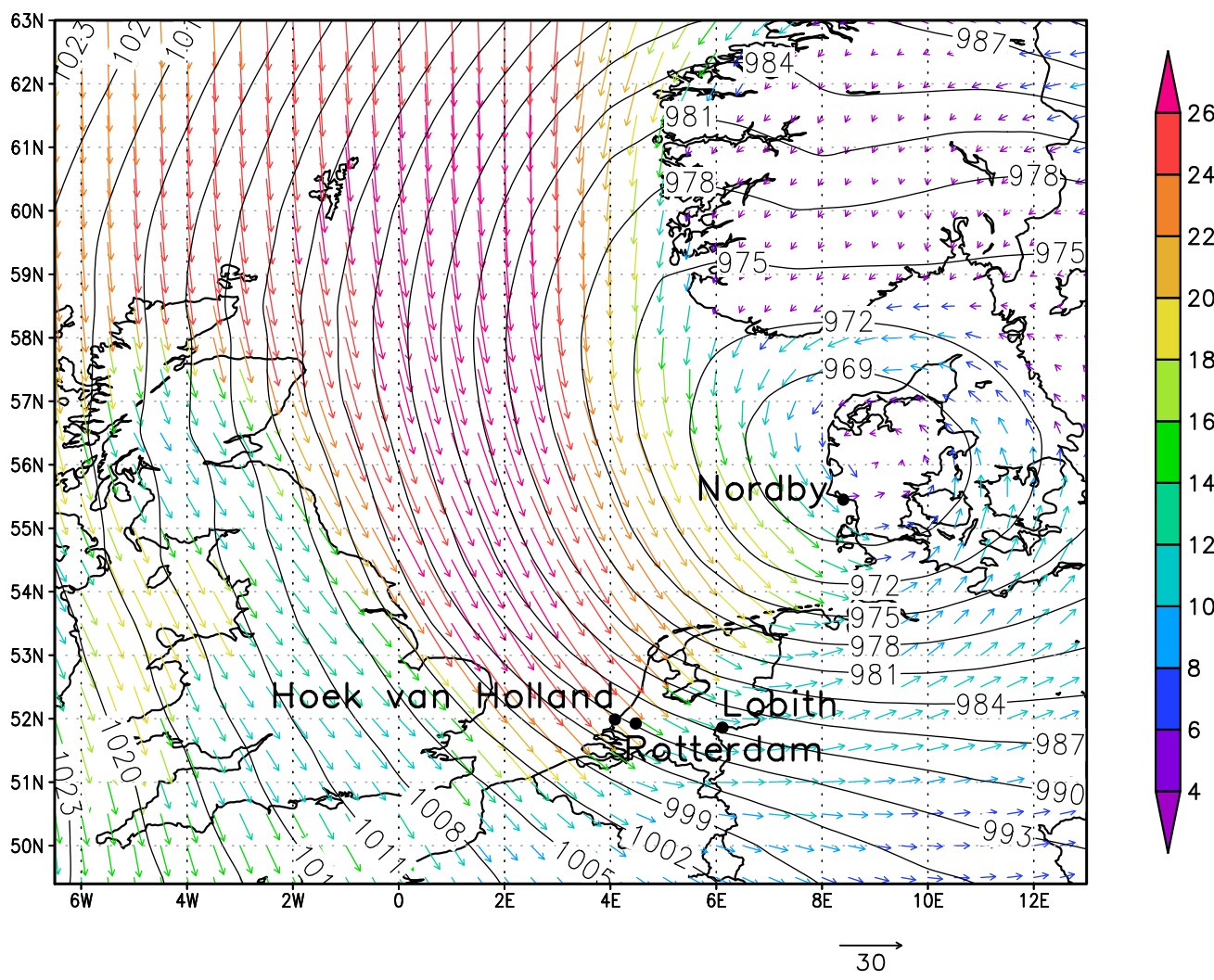

**Figure 4.** Wind and pressure fields for the situation leading to the highest surge in Hoek van Holland that occurred in the ECMWF-WAQUA/DCSMv5 ensemble. The locations of Hoek van Holland, Rotterdam, Lobith and Nordby are indicated.

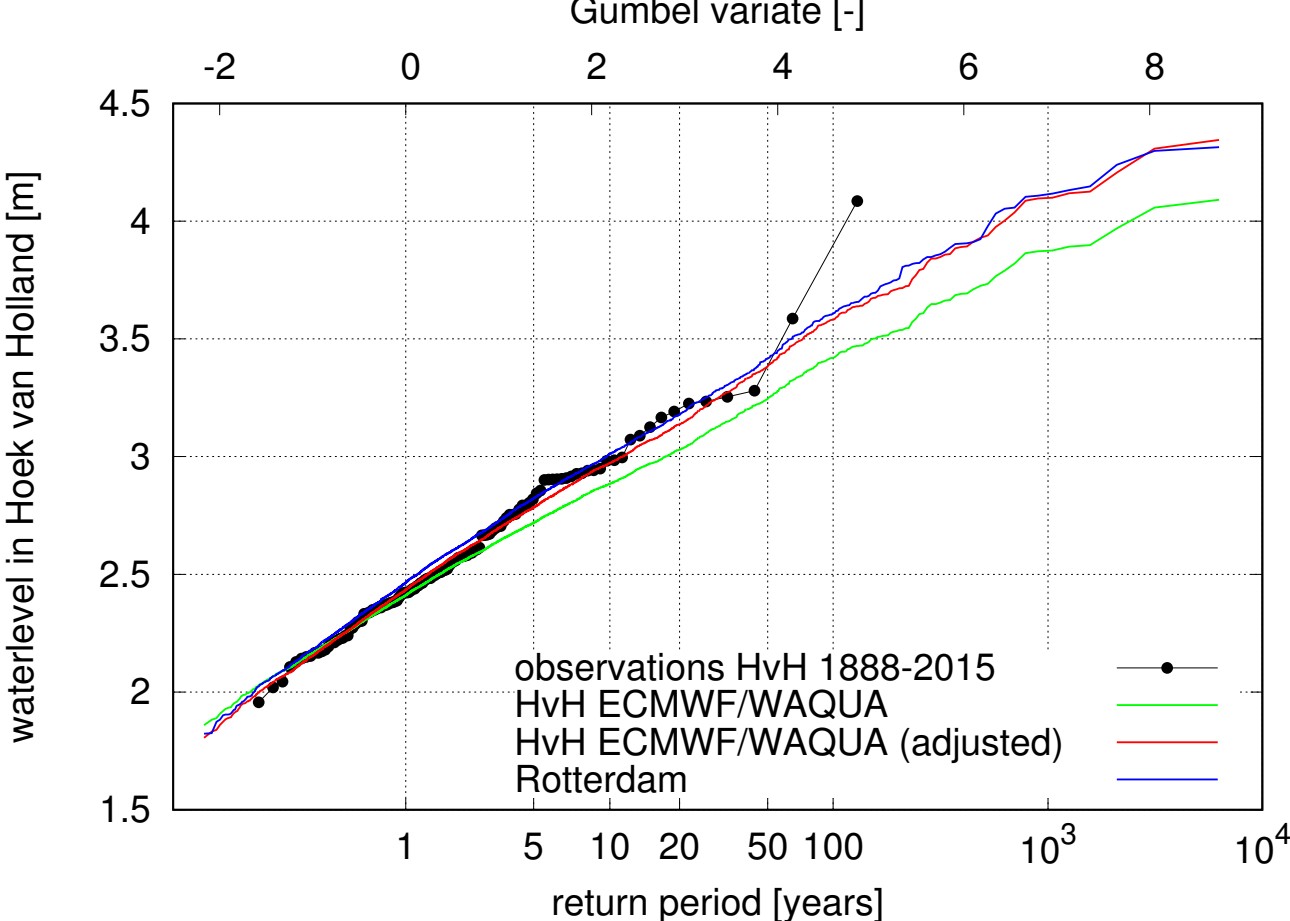

**Figure 5.** Gumbel plot of the annual maximum water levels in Hoek van Holland according to the observations (black) and the ECMWF-WAQUA/DCSMv5 ensemble (red). Adjusting the Gumbel parameters of the ECMWF-WAQUA/DCSMv5 ensemble to match the observed Gumbel parameters (according to Eq (10)) results in the blue distribution.

In order to correct for this feature, we applied the following correction to the ECMWF-WAQUA/DCSMv5 water levels at Hoek van Holland:

$$L_{adj} = \mu_{obs} + \sigma_{obs} \frac{L_{org} - \mu_{EW}}{\sigma_{EW}} \tag{10}$$

5   in which $L_{org}$ is the original water level as calculated by ECMWF-WAQUA/DCSMv5, and $L_{adj}$ the adjusted water level. The subscripts *obs* and *EW* refer to the Gumbel parameters of the observations and ECMWF-WAQUA/DCSMv5, respectively. The quantile mapping of Eq (10) ensures that the ECMWF-WAQUA/DCSMv5 water levels have the same Gumbel location and scale parameter as the observations have. The results presented in this paper are based on $L_{adj}$.

Correction according to Eq (10) results in the red line of Figure 5. For the once-in-10-years return level this implies a correction of only 3% in the water level, and 5% in the surge (taking the average astronomical high tide of 1.21 m into account).

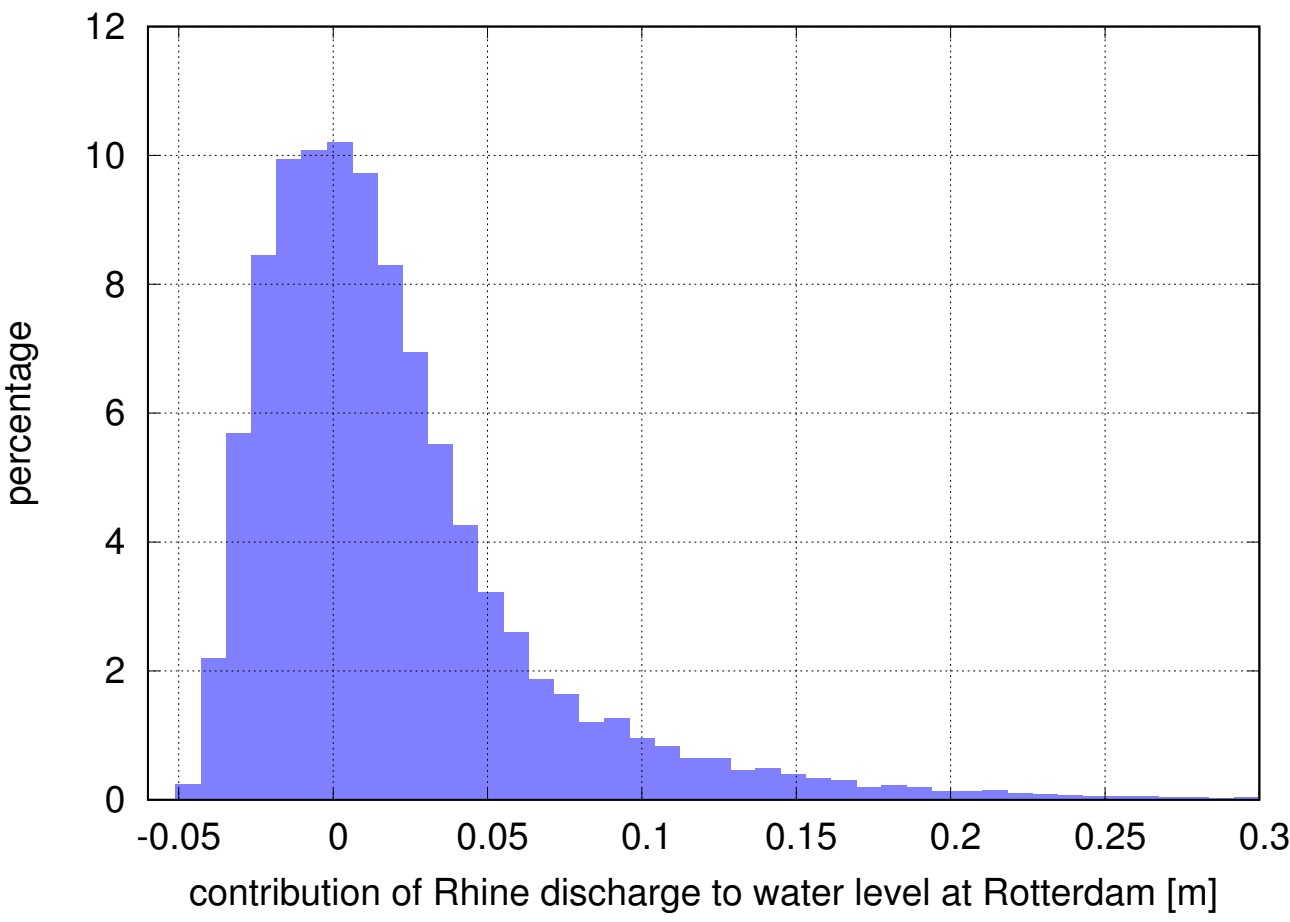

**Figure 6.** Histogram of the effect of the Rhine discharge on the water level in Rotterdam. The bin width is 0.5 cm.

We conclude that, although a correction is necessary, this correction is small enough to trust the water levels of the ECMWF-WAQUA/DCSMv5 ensemble for determining the closure frequencies.

### 4.3 Rhine discharge

5   Figure 6 shows the histogram of the effect of the Rhine discharge on the water level at Rotterdam.

In the observational record the maximum effect of the river discharge to the water level in Rotterdam (according to Eq (1)) is 0.43 m for the highest observed discharge of 12,280 m$^3$/s at Lobith. The yearly averaged addition of the river discharge to the water level in Rotterdam is 0.02 m, and the average effect to the annual maxima at Rotterdam is 0.03 m. This means that the Gumbel plot of the annual maximum water levels at Rotterdam is about 0.03 m higher than that of Hoek van Holland (see

10   blue line in Figure 5). We note that the equation used by Zhong et al. (2012) to model the effect of the Rhine discharge on the water level in Rotterdam gives identical results.

We conclude that the effect of the Rhine discharge on the water level in Rotterdam can be substantial, but that the average effect on the extreme levels is only a few centimeters.

## 5 Results

Figure 7 shows the distribution of the inter-arrival times for two thresholds: 2 m (upper panel) and 3 m (lower panel). The insets show the distribution for the first 14 and 30 days, respectively. Note that the unit of the horizontal axis of the upper panel is in days, and of the lower panel in years. Figure 7 illustrates the following items: First, in the 6282-year dataset, the maximum recurrence time for a level of 2 m is 1095 days, and 135 years for a level of 3 m. Second, the deviations from the straight black line (which represents an exponential distribution on the logarithmic vertical scale) indicate that the distribution of recurrence times is not a Poisson process - neither for the 2 m nor for the 3 m threshold. Especially in the upper graph the seasonal variation in the recurrence times is clearly visible by the oscillation around the black line. This is the result of the low probability in summer and higher probability in winter that a 2 m event occurs. Thirdly, as illustrated by the insets, the probability of a recurrence time of less than 5 days is considerably higher than independence would indicate. Apparently, there is clustering of extreme events up to inter-arrival times of 5 days. This is in agreement with Mailier et al. (2006) who quantify the clustering of extra-tropical cyclones for the area of interest. Also the influence of spring tide will increase the probability of double closures (Van den Hurk et al., 2015). Fourthly, the inset of the upper panel also shows the influence of the deterministic astronomical tide: the figure shows a 12.5-hourly oscillation, caused by the fact that all exceedences of the thresholds occur at high tide. Fifth, the inter-arrival times for very high thresholds (as shown in the lower panel for the 3 m threshold) are distributed according to Equation 7, which implies that the inter-arrival times at annual scale can be considered to be independent[3]. Sixth, the ECMWF-WAQUA/DCSMv5 shows a good agreement with the observations for the 2 m threshold (blue line in upper panel). As there only 10 exceedences above 3 m threshold in the observational record (blue line in lower panel), it is hard to verify the distribution for the 3 m threshold. None of the 3 m exceedences in the observations are within a month of each other.

From Figure 7 it can be concluded that the assumption of independence for the occurrence of (extreme) water levels is violated on a daily scale by the astronomical tide, on a weekly scale by the clustering of extra-tropical cyclones and spring tide, and on monthly scale by the seasonal variation in the storm intensity and frequency. Only at annual scale the inter-arrival times are exponentially distributed, and thus can be considered to be independent. This means that we cannot assume independence in order to calculate recurrence times for short intervals, and thus cannot apply Equation 7 to estimate the probability of a double closure within a week or month.

Instead of assuming independence, we could directly count the inter-arrival times between the events that exceed 3 m water level, and construct an empirical distribution function (EDF) from them (see section 3.3). However, even the 6282-year dataset is too short for this approach, as the dataset contains only 30 inter-arrival times that are less than a month. Direct derivation of the EDF is therefore not possible.

---

[3]Although the forecasts runs are only 7 months in length, they are combined in such a way that a possible oceanic influence remains present, see Table 1.

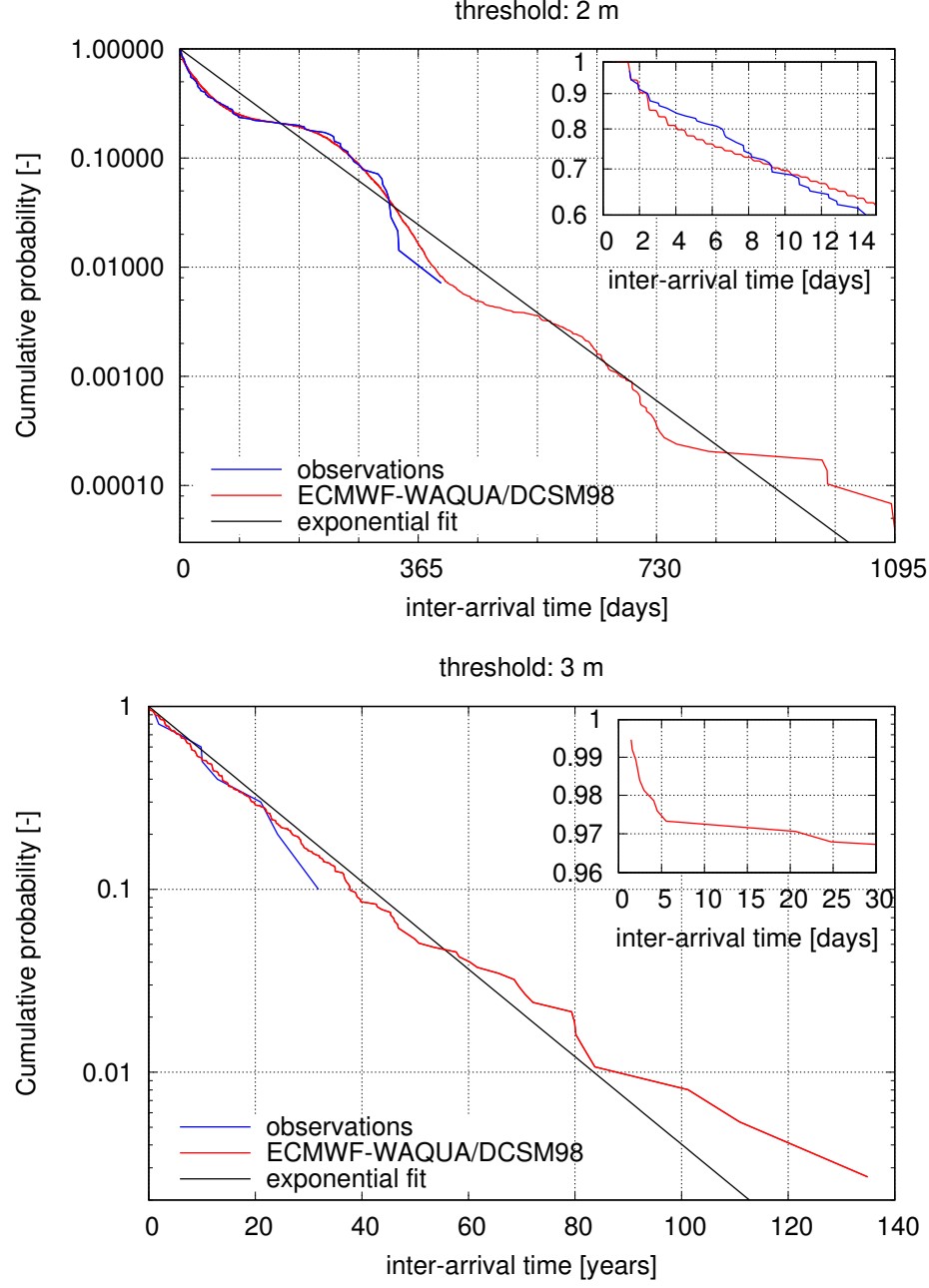

**Figure 7.** Distributions of the recurrence times of the water level in Hoek van Holland for a threshold of 2 m (upper panel) and 3 m (lower panel). The insets show the distribution for the first 14 and 30 days, respectively. The black line represents an exponential fit to the whole dataset. The vertical axes are logarithmic. The blue lines represent the observations.

In order to bypass this problem, we explore how the required EDF for 3 m relates to the EDF for lower thresholds. Figure 8 shows that the number of occurrences in which the threshold is exceeded twice within 1 week is exponentially related to the threshold (blue line and points). The same holds for inter-arrival times of 2 and 4 weeks (green and red resp.).

Fitting the line

$$\ln(N) = N_0 - y/\beta \tag{11}$$

where $N$ is the count, $y$ the threshold and $\beta$ the slope, yields a value of $N_0$ that depends on the time window, and $\beta = 0.145 \pm 2\%$ for all three lines.

The figure indicates that we can base our desired EDF on a lower threshold than 3 m, and transform those results to the
required EDF for a threshold of 3 m by a simple multiplication. The value of that multiplication factor $M$ for the probabilities of the 3 m threshold is given by:

$$M = \frac{N_1}{N_2} = \exp(\frac{y_2 - y_1}{\beta}) \tag{12}$$

in which $N_1$ is the counted number of occurrences for which the water level exceeds $y_1$ m twice within the given time window, and $N_2$ the number of double closures for level $y_2$ (3.0 m) we are looking for. The fact that the three lines in Figure 8 are
parallel indicates that this multiplication factor is virtually independent of the time window.

We chose to derive the EDF on a threshold of $y_1$ equal to 2.5 m, as this threshold gives a good compromise between the number of occurrences (we then have 1228 events that occur within 4 weeks of the previous event, and 601 events that occur within 1 week) and the extremity of the threshold (the 2.5 m threshold is exceeded on average once in two years, see Figure 5). According to Eq (12), the probabilities of the 2.5 m threshold have then to be multiplied by 0.032 to be transformed to the
3.0 m threshold.

## 5.1   Recurrence times

Figure 9 shows the recurrence times as a function of the inter-arrival times, for a threshold of 3 m.

The green shading in the figure illustrates the effect on the recurrence time if the EDF is based on thresholds in the range of 2.3-2.7 m. Apparently, the outcome varies about 10% with the choice of the threshold. The blue shading represents the
uncertainty due to the bias correction of Eq 10, in which $\mu_{obs}=2.43$ and $\sigma_{obs}=0.264$ are replaced by $\mu \pm 2\Delta\mu_{obs}$ (2.38,2.47) and $\sigma \pm 2\Delta\sigma_{obs}$ (0.23,0.30), respectively. The figure shows that the uncertainty due the bias correction is much larger than due to the threshold selection.

From Figure 9 it can be seen that in the current climate once in about 150 (70,390) years, the Maeslant-barrier has to be closed twice in a month due to exceedence of the 3.0 m threshold. Once in about 330 (150,810) years it has to be closed twice
in a week. Here, the number between brackets indicate the 95% uncertainty ranges based on the bias correction, showing that the uncertainty in the recurrence time is slightly more than a factor of two.

The oscillations in the graph of Figure 9 are caused by the fact that exceedences of the threshold always occur at high tide, i.e. the inter-arrival times are always a multiple of 12.5 hours.

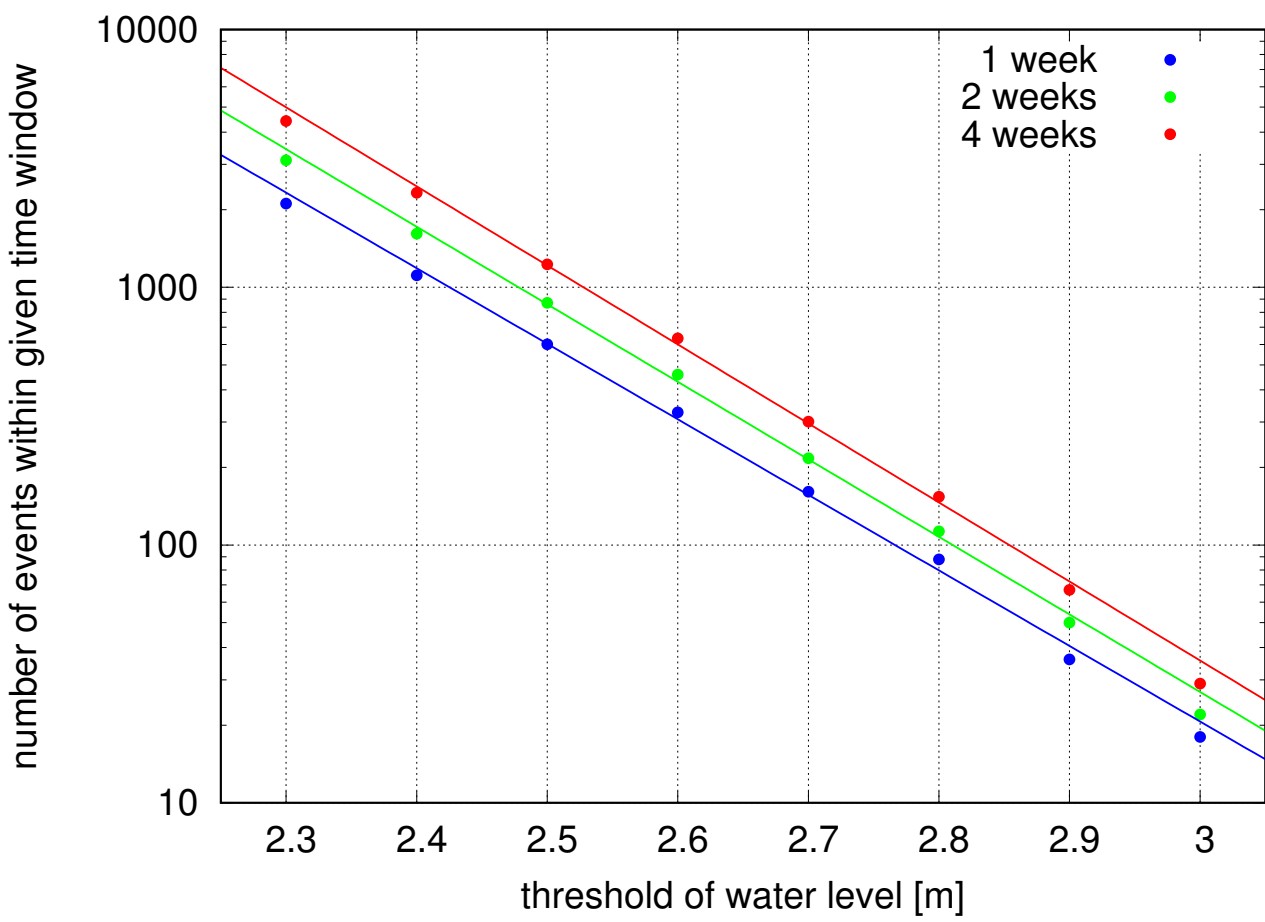

**Figure 8.** Number of occurrences $N$ in which the threshold $y$ is exceeded twice within 1, 2 or 4 weeks (blue, green and red respectively). The vertical axis is logarithmic.

The dashed line represents the Poisson distribution (Eq 7) with $\lambda = 0.10$ per year. It shows that the assumption of independence leads to considerable deviations in the estimation of the recurrence times of double events.

### 5.2 Effect of sea level rise

5  In order to estimate the first-order effect of sea level rise on the closure frequency, we assume no changes in the wind climate, no change in river discharge, and no effect of the sea level rise on the surge and astronomical tides (which is approximately true, see e.g. Lowe et al., 2001; Sterl et al., 2009). In that case, the effect of sea level rise can be incorporated by considering the probabilities of a threshold that is accordingly lower. A sea level rise of 0.3 m will thus lead to the situation as if closure takes place at 2.7 m instead of 3.0 m.

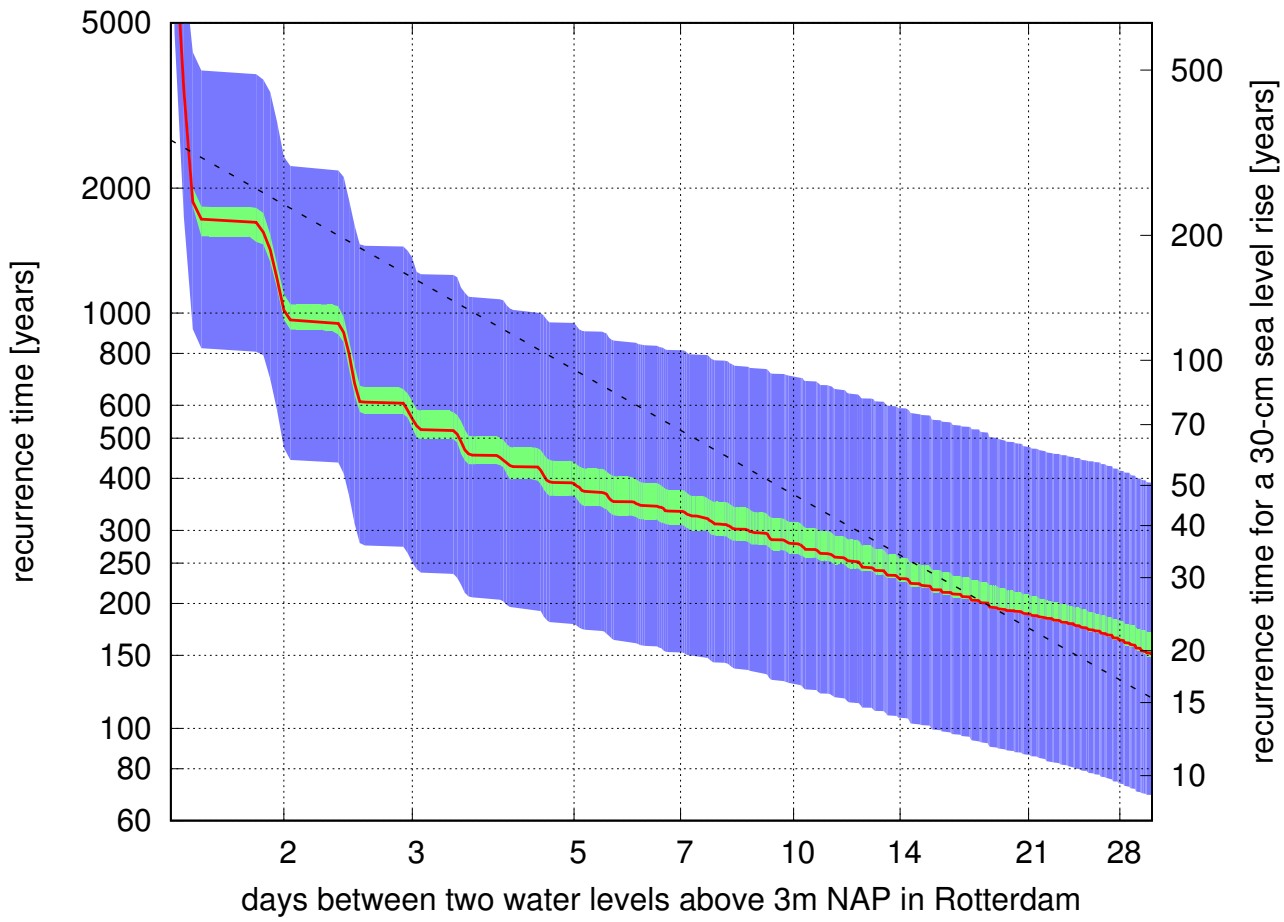

**Figure 9.** Recurrence times in years as a function of the inter-arrival time for a threshold of 3.0 m. The line is determined by counting the events for a threshold of 2.5 m and multiplying the probabilities with 0.032 (according to Eq (12).). The green shading illustrates the effect on the recurrence time if the EDF is based on thresholds in the range of 2.3-2.7 m. The blue shading is the 95% confidence interval due to the bias correction of Eq 10. The dashed line represents the Poisson distribution (Eq 7) with $\lambda = 0.10$ per year. The right axis shows the recurrence times for a sea level rise of 30 cm. All axes are logarithmic.

### 5.2.1    Effect of sea level rise on single closures

The effect of the sea level rise on the number of single closures can be derived by calculating $\partial T / \partial y$ from Eq (5). It easily follows that

$$5 \quad \ln(\frac{T_{s,2}}{T_{s,1}}) \approx \frac{y_2 - y_1}{\sigma} \tag{13}$$

in which return periods $T_{s,1}$ and $T_{s,2}$ belong to water levels $y_1$ and $y_2$, respectively. Here, $\sigma$=0.26 is the Gumbel scale parameter. It directly follows from Eq (13) that a 0.18 m sea level rise doubles the closure frequency. With the expected sea level rise of

0.15-0.40 m in 2050 with respect to 1981-2010 (Van den Hurk et al., 2014), the closure frequency will increase by a factor of 1.8-4.6.

### 5.2.2 Effect of sea level rise on double closures

The effect of sea level rise on the probability of two closures within a time window can directly be derived from Figure 8 and Equation 12. In a way similar to Eq (13) it follows from Eq (11) that

$$\ln\left(\frac{N_1}{N_2}\right) = \frac{y_2 - y_1}{\beta} \tag{14}$$

Equation (14) shows that approximately every 0.10 m sea level rise doubles the probability that two closures occur within a given time window. The expected sea level rise of 0.15-0.40 m in 2050 results in recurrence times that are 2.8-16 times more frequent than in the reference situation. These recurrence times for 0.3 m sea level rise are indicated on the right axis of Figure 9.

## 6 Conclusions

The seasonal forecasts of the ECMWF, with a total length of more than 6000 years, represent the current wind climatology over the North sea area very accurately. Combination of the ECMWF output with the surge model WAQUA/DCSMv5 results in a 6282-year dataset of water levels that (after a small correction) are well suited for many research objectives.

In this paper we apply the dataset in order to estimate how often the movable Maeslant-barrier in the New Waterway (which is the artificial mouth of the river Rhine) has to be closed twice in a short time interval - varying between days up till a month. This is of importance as the barrier might get damaged during the first closure and the barrier can not be closed during the reparation time.

Assuming independence between two closures leads to wrong estimates of the double closures. Independence is violated by the deterministic component of the astronomical tide on the daily scale, by clustering of depressions and by spring tide on the weekly scale, and by seasonality on the monthly scale.

By counting the number of double events over a threshold of 2.5 m, and using that the number of events is exponentially related to the threshold, it is found that the barrier has to be closed within a month approximately once in 150 years, and once in 330 years within a week. The large uncertainty in these recurrence intervals of more than a factor of two are caused by the sensitivity of the results to the Gumbel parameters of the observed record, which are used for bias correction.

Sea level rise has a large impact on the frequency of single and double closures. Every 10 cm sea level rise doubles the probability of double closures, resulting in 2.7-14 times more double closures for the 0.15-0.40 m expected sea level rise in 2050.

*Acknowledgements.* This work was initiated and funded by Rijkswaterstaat.

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
