# Peer review of "Recurrence intervals for the closure of the Dutch Maeslant surge"

_Ocean Science, 2017_

## Referee Comment (RC1) · Anonymous Referee #1 · 25 May 2017

General comments

The authors, van den Brink and de Goederen, address an issue of high importance regarding coastal risk in Rotterdam, namely the estimate of the chance that a storm occurs during the maintenance/repair of the movable surge barrier New Waterway, which protects Rotterdam city. The authors propose to estimate the frequency of succeeding closures within a given interval (e.g., one day, one month), i.e. the inter-arrival times. Due to the short instrumental water level time series, the authors propose to rely on a numericcaly calculated water level series derived from the combination of ECMWF forecasts with the WAQUA/DCSMv5 surge model. An evaluation of the effect related to sea level rise is also performed.

The manuscript is well written; the presentation of the problematic, the methods and

results are clear as well as the figures. However, my main concern is regarding the assumptions made by the auhors. There are several aspects that need either to be clarified or better presented or discussed with respect to alternative methods in the litterature. That is why, I recommend revisions before publication.

Specific comments

Instrumental observations The authors underline from the introduction (from line 26), with reasons, that the use of ECMWF derived time series is a real plus regarding the use of instrumental observations alone. Yet, it would beneficial regarding the readability of the paper to have from the start a clear vision of what the instrumental time series is (duration, quality, what would be the estimate of inter-arrival time using it alone, etc.). Furthermore, adding more details regarding the way the correcction due to land subsidence / long term trend (Figure 1) has been derived would also be appreciated (Sect. 4.1).

Presentation of the methodology In section 3, the authors present the methods used but it would be beneficial to have a section recalling the different stages of their whole strategy, because the link between the methods is not clear. As far as I understand, the authors only perform Extreme Value Analysis to correct the bias of ECMWF derived time series and to be used to fit the exponential model for the inter-arrival times. Besides in the section 5, which is dedicated to result analysis, the authors present an another method, which is confusing.

Alternative approaches - The statistics of inter-event spacing can also be analysed using the tools of extreme value analysis as it has been done regarding extreme beach erosion; for instance Callaghan et al. (2008): Coastal Engineering. Could the authors discuss/ comments on how their approach differ / complement? - The extreme value analysis is performed using a GEV distribution. Could the authors justify their choices, especially regarding the use of a more complete approach with GPD distribution. - As stated by the authors in Section 2.3 with references to van den Brinck et al. 2005A,

Kew et al. 2013), the joint occurrence of extreme discharge and extreme water levels is independent. Could the authors comment on the manner to address the problem when this situation does not exist.

Results - The authors seem to base their approach on the independence assumption. In section 3 and in the introduction, the authors underline the use of the Poisson process model, but in section 5, they underline the violation of the assumption, and propose an alternative. What is confusing is that this alternative is not highlighted in the abstract neither in the conclusion (reading line 13 'assuming independence...' leads to wonder what can be done to overcome the independence problem). - Regarding the presentation of the results (figure 7 and 8 in particular), could the authors discuss / comment on the uncertainty related to the use of an empirical probability distribution. More specifically, there is some slight scattering of the dots in Figure 7, which makes wonder about the uncertainty in the results described in Sect. 5.1. - The question of uncertainty is also raised regarding the GEV fit (Fig. 4) and the derived bias correction.

Technical corrections Figure 6: in the legend, the blue line for observations appears twice.

---

## Referee Comment (RC2) · Anonymous Referee #2 · 31 May 2017

The paper addresses the assessment of risks associated with high water extreme events reoccurring within a certain period of time with the practical application to the potential surge barrier closures of the Rotterdam Maeslant barrier. The extreme event statistics is constructed based on the seasonal ECMWF forecast data, which provides a set of physically plausible situations and can be considered as a quasi-observational dataset of several thousands years. Additionally, river discharge is approximated for Rotterdam from the 100 years of observations. Extreme value distribution is fitted based on annual maxima of the water level data to estimate the frequency of occurrence. The idea of estimating events inter-arrival times by Poisson process is proposed and rejected due to violation of the independency requirement. Instead, the empirical probability distribution function for lower thresholds is constructed from the dataset and then extrapolated to higher thresholds. It is also shown how linear consideration of sea

level rise would influence the double closure frequencies within the established model frame.

The paper is well structured; the goals and results are clearly described. The discussed problematic is relevant and is of both methodological and resultant interest. However, the methodology description and its applications are not always straightforward. In particular, the part concerning Poisson process (section 3.2 and further on) is misleading because, as far as I understood, it is used only to show the falseness of the independency assumption. Presumably, the trial of this method was part of the research process and search for the proper estimate of the reoccurring events. In this case the description of the method seems to be superfluous for this paper.

The usage of forecast members is a valid and effective approach to expand the dataset in the lack of instrumental measurements. Just for understanding, did the authors have 2 model simulations - with and without meteo-forcing, or astronomical tides were analytically estimated (p3. lines 32-33)? Are surge heights used somewhere in this study? It looks like all the data analysis is done based on water level timeseries, in this case, please remove 'surge' from the text (e.g. p3. line 18).

There are several sources of uncertainty emerging throughout the analysis; it would be helpful to see the estimate of total uncertainty range. This is partly done in Figure 8 and section 5.1, but what about the uncertainties from GEV estimate and correction? In the conclusion (lines 40-42) very precise numbers are given without any potential error intervals, an additional sentence or two and a rough estimate in percentage of the total results would suffice here.

Minor comments:

- in abstract, p8 line 9 and maybe somewhere else in the text: reduplicates is presumably used in the meaning of "a factor of two" but cannot be used in this sense and is misleading in the context. Please use other synonyms (doubles, redoubles, duplicates,. . .)

- Eq.5 and others use log and p.3 line80 use ln. If log is natural logarithm, please use the same notations everywhere.

- p5. lines 13: L_adj is actually adjusted surge or rather adjusted water level?

- p3. line 77: please coordinate singular/plural forms The distribution(s) . . . are (is). . .

- fig.1 and 4.1: is the correction made by adding these values to the observed data for each year? Where the numbers are coming from?

- p5. line 33: how does the value 0.57m came from the eq. (1) using 12280mˆ3/s discharge I come out with 0.43m. . .

- p5. line 38-39: it is not that the water levels at Rotterdam are 3cm higher than at Hoek van Holland, it is that for Rotterdam you consider additionally the river discharge and for HvH only the sea level. Please reformulate the sentence, it is misleading.

- p5. line 27-28: 'River run-off is not considered in this paper'. Do the authors mean it is not considered in the hydrodynamic model? Because later in the paper there is a talk about run-off again.

- fig.6: upper panel – 2 times "observation" In the legend.

---

## Author Comment (AC1) · 11 Jul 2017

Below we reply to the comments of anonymous referee #1. The original comments are given in *italics*.

- *Instrumental observations*

  - *The authors underline from the introduction (from line 26), with reasons, that the use of ECMWF derived time series is a real plus regarding the use of instrumental observations alone. Yet, it would beneficial regarding the readability of the paper to have from the start a clear vision of what the instrumental time series is (duration, quality, what would be the estimate of*

[Figure]

*inter-arrival time using it alone, etc.).*

We agree with the referee that the section about the observations is too short. In order to improve the readibility, we did the following:

1. we renamed the section "Models" to "Models and observations".
2. we moved the section 4.1 ("Water level observations") to section 2.4.
3. We extended the section about the Water level observations (2.4) with extra information about the duration and quality as follows: "The observational record of water levels at Hoek van Holland starts in 1864 (Holgate et al, 2013; PSMSL, 2017). Accurate readings of the water level start in August 1887. We used the data from 1888 onward. The data before 1987 are obtained visually from (digitized) charts, afterwards 10-min average values are used." Additionally, we added an extra figure with the timeseries of the observed and corrected annual maxima in Hoek van Holland, with the following accompanying text: "The annual maxima of the water level in Hoek van Holland are shown in Figure 2, both uncorrected and corrected. The observational record contains (after correction) 10 events that exceed 3 m in Hoek van Holland, the smallest interarrival time being 1.2 years (in 1953 and 1954). This makes direct derivation of the recurrence intervals of interarrival times smaller than 1 month impossible."

– *Furthermore, adding more details regarding the way the correction due to land subsidence / long term trend (Figure 1) has been derived would also be appreciated (Sect. 4.1).*

(Note that section 4.1 has been moved to section 2.4.) The analysis has been done in the 'Basispeilen langs de Nederlandse kust' report (1993), which can be found online at https://repository.tudelft.nl/islandora/object/uuid:c5fb1012-e296-49cf-a6fa-95d4ed58000c/datastream/OBJ/download, (page 56). It contains an analysis of the discontinuities and trends. A discontinuity in 1965 has been

observed, which is attributed to the extension of the Rotterdam harbour to the west (as mentioned in the manuscript). The trend is determined separately before 1965 and after 1965. We re-calculated the trend after 1965, because of the extension of the observational record with 30 years (1986-2015).

We added the reference to the Basispeilen-report to the manuscript. The reason for not referencing it in the earlier version is that it is written in Dutch.

- *Presentation of the methodology*
  *In section 3, the authors present the methods used but it would be beneficial to have a section recalling the different stages of their whole strategy, because the link between the methods is not clear. As far as I understand, the authors only perform Extreme Value Analysis to correct the bias of ECMWF derived time series and to be used to fit the exponential model for the inter-arrival times. Besides in the section 5, which is dedicated to result analysis, the authors present another method, which is confusing.*
  We added an introductory part to Section 3, in which we introduce the different parts of the analysis, and their mutual relationship. Additionally, we added a small section about the Empirical Distribution Function (section 3.3).

- *Alternative approaches*

  - *The statistics of inter-event spacing can also be analysed using the tools of extreme value analysis as it has been done regarding extreme beach erosion; for instance Callaghan et al. (2008): Coastal Engineering. Could the authors discuss/ comments on how their approach differ / complement?*
    Callaghan et al. (2008) do model the seasonal variation of the event occurrence. However, they do not model the event-clustering explicitly (p 382).
    Although Callaghan et al. (2008) uses advanced techniques to model many

processes, the fact that all processes have to be modelled statistically (and the required parameters have to be estimated from the short dataset) is at the same time the weakness of their approach, especially because these models are extrapolated outside the range of the observations.

In our approach, the results are much less sensitive to the (parameters of the) statistical models than the results of Callaghan. On the other hand, our approach heavily relies on the quality of the both the meteorological and hydrological model, whereas Callaghan uses observations.

In our opinion, the combination of the ECMWF model with the WAQUA/DCSMv5 outperforms the statistical approach of Callaghan, especially in situations where observational records are either lacking, short or of bad quality.

– *The extreme value analysis is performed using a GEV distribution. Could the authors justify their choices, especially regarding the use of a more complete approach with GPD distribution.*
It is, in our opinion, unlikely that the GPD will give better results than the GEV distribution, as both approaches have 3 parameters. The GPD might results in better estimates for short record lengths, as it allows multiple (independent) events per year to be considered in the estimation of the parameters. However, in our case, this will not be of any importance for the 6282-year ECMWF record, and also for the 128-year observational record of Hoek van Holland, the application of the GPD will not lead to (significantly) different results than the GEV does.

– *As stated by the authors in Section 2.3 with references to Van den Brink et al. (2005), Kew et al. (2013), the joint occurrence of extreme discharge and extreme water levels is independent. Could the authors comment on the manner to address the problem when this situation does not exist.*
We added the following sentence as a footnote to the manuscript (section 2.3): "In the case that extreme discharge and extreme water levels are correlated, the most promising solution - in line with the topic of this paper - is to use the precipitation amounts, the temperature and snow melt in the Rhine basin as input for a hydrological model to calculate the Rhine discharge. In this way no explicit assumptions about the correlation have to be made. This is however outside the scope of this paper."

- *Results*

  – *The authors seem to base their approach on the independence assumption. In section 3 and in the introduction, the authors underline the use of the Poisson process model, but in section 5, they underline the violation of the assumption, and propose an alternative. What is confusing is that this alternative is not highlighted in the abstract neither in the conclusion (reading line 13 'assuming independence...' leads to wonder what can be done to overcome the independence problem.*
  We added the following sentence to the abstract: "We show that the Poisson process model leads to wrong results, as it neglects the temporal correlations that are present on daily, weekly and monthly scales. By counting the number of double events over a threshold of 2.5 m, and using that the number of events is exponentially related to the threshold, . . . " We also reshuffled the Conclusions, by first mentioning that the assumption of independence is violated, and afterwards that we used the empirical distribution to count the number of double closures.

  – *Regarding the presentation of the results (figure 7 and 8 in particular), could the authors discuss / comment on the uncertainty related to the use of an empirical probability distribution. More specifically, there is some slight scattering of the dots in Figure 7, which makes wonder about the uncertainty in the results described in Sect. 5.1*
In our opinion, we have adressed this point by indicating the effect when another threshold is used than 2.5 m. This effect on the recurrence times is indicated by the blue shading in figure 9 (old figure 8). Note that the effect of the uncertainty in the Gumbel parameters of the observed record (which is added based on the question below) is considerably larger than the uncertainty due to the choice of the threshold. Both uncertainties are indicated in Figure 9 (old figure 8).

– *The question of uncertainty is also raised regarding the GEV fit (Fig. 4) and the derived bias correction.*
We admit that neglecting this uncertainty was a shortcoming of the manuscript. We thank the referee for pointing this out.
Neglecting the uncertainty in the GEV fit of the 6282-year ECMWF-WAQUA/DCSMv5 dataset, we focus on the uncertainty in the Gumbel fit of the observations. A first-order estimation of the 95% uncertainty range is made by replacing $\mu_{obs}$ and $\sigma_{obs}$ in Equation 9 with $\mu_{obs} \pm 2\Delta\mu_{obs}$ $\sigma_{obs} \pm 2\Delta\sigma_{obs}$, respectively. Redoing the calculations with those ajusted bias corrections gives a good indication of the uncertainty range in the estimated recurrence intervals. We have added these intervals to Figure 9 (old figure 8). It indicates that this uncertainty due to the statistical uncertainty in $\mu_{obs}$ and $\sigma_{obs}$ is much larger than the uncertainty due to the choice of the threshold.
We added an extra paragraph about the uncertainty (section 3.4) in which we explain how we derived the uncertainty in figure 9 (old figure 8).

- *Technical corrections*

    – *Figure 6: in the legend, the blue line for observations appears twice.*
    corrected

---

## Author Comment (AC2) · 11 Jul 2017

Below we reply to the comments of anonymous referee #2. The original comments are given in *italics*.

**0.1 Major comments**

- *The methodology description and its applications are not always straightforward. In particular, the part concerning Poisson process (section 3.2 and further on) is misleading because, as far as I understood, it is used only to show the falseness of the independency assumption. Presumably, the trial of this method was part of the research process and search for the proper estimate of the reoccurring*

[Figure]

*events. In this case the description of the method seems to be superfluous for this paper.*

The same point is mentioned by referee # 1. We agree with the referees that the description was not clear in all points. We changed the manuscript as follows:

– We added the following sentence to the Abstract: "We show that the Poisson process model leads to wrong results, as it neglects the temporal correlations that are present on daily, weekly and monthly scales. By counting the number of double events over a threshold of 2.5 m, and using that the number of events is exponentially related to the threshold, . . . "

– We added an introductory paragraph to the Methodology section, in which we explain that the Poisson process model fails.

– We reshuffled the Conclusions, by first mentioning that the assumption of independence is violated, and afterwards that we used the empirical distribution to count the number of double closures.

• *The usage of forecast members is a valid and effective approach to expand the dataset in the lack of instrumental measurements. Just for understanding, did the authors have 2 model simulations - with and without meteo-forcing, or astronomical tides were analytically estimated (p3. lines 32-33)?*

The WAQUA-DCSMv5 has two options considering the astronomical tides. The first one is to use the tide that the model calculates, given the astronomical input from the boundaries. This result is indeed obtained by running the model without meteorogical forcing. The second option is to use tidal constituents calculated from observations for specific locations.

We used the first option, i.e. the tide as calculated by WAQUA-DCSMv5.

• *Are surge heights used somewhere in this study? It looks like all the data analysis is done based on water level timeseries, in this case, please remove 'surge' from the text (e.g. p3. line 18).*

We agree with the referee that the use of the word 'surge' is misleading. We replaced the word with 'water level'.

- *There are several sources of uncertainty emerging throughout the analysis; it would be helpful to see the estimate of total uncertainty range. This is partly done in Figure 8 and section 5.1, but what about the uncertainties from GEV estimate and correction? In the conclusion (lines 40-42) very precise numbers are given without any potential error intervals, an additional sentence or two and a rough estimate in percentage of the total results would suffice here.*
  We agree with the referee(s) that the discussion about the uncertainties was insufficient. We added a discussion about the different uncertainties as follows:

  – We added an extra paragraph about uncertainty analysis (Section 3.4) in which we discuss two sources of uncertainty that the referee mentions, i.e. due to bias correction and due to the choice of the threshold.

  – We added the estimate of the 95% uncertainty range due to the bias correction in Figure 9 (old Figure 8). The figure shows that the uncertainty in the bias correction is considerably larger than the uncertainty due to the choice of the threshold. We thank the referee for pointing this out.

  – We added the uncertainty ranges also to the Abstract, Results and Conclusions.

**0.2 Minor comments**

- *in abstract, p8 line 9 and maybe somewhere else in the text: reduplicates is presumably used in the meaning of "a factor of two" but cannot be used in this sense and is misleading in the context. Please use other synonyms (doubles, redoubles, duplicates,. . .)*
  We changed the word "reduplicates" with "doubles"

- *Eq.5 and others use $\log$ and p.3 line80 use $\ln$. If $\log$ is natural logarithm, please use the same notations everywhere.*
  We changed $\log$ to $\ln$.

- *p5. lines 13: $L_{adj}$ is actually adjusted surge or rather adjusted water level?*
  This should be (adjusted) water level. We changed it accordingly.

- *p3. line 77: please coordinate singular/plural forms The distribution(s) . . . are (is) . . .*
  We changed it to plural.

- *fig.1 and 4.1: is the correction made by adding these values to the observed data for each year? Where the numbers are coming from?*
  Indeed the correction is made by adding the corrections to every year (rounded to cm). The correction is presented in the 'Basispeilen'-report (1993), which is added to the references.

- *p5. line 33: how does the value 0.57 m came from the eq. (1) using 12280 $m^3$/s discharge I come out with 0.43 m*
  The referee is (of course) right, the number should be 0.43 m. Fortunately, the calculations are done with the correct formula. We adjusted the number to 0.43.

- *p5. line 38-39: it is not that the water levels at Rotterdam are 3cm higher than at Hoek van Holland, it is that for Rotterdam you consider additionally the river discharge and for HvH only the sea level. Please reformulate the sentence, it is misleading.*
  We agree that the WAQUA-DCSMv5 model does not take the river discharge into account in calculating the water level at Hoek van Holland, whereas the observations at Hoek van Holland are influenced by the river discharge. However, from De Goederen (2013, page 16) it can be derived that the effect of the river discharge on Hoek van Holland for the maximally observed discharge (12280 $m^3$/s)

is only 4 cm, i.e. ≈10% of the effect in Rotterdam. The effect on the Gumbel location parameter is only 8 mm. It is therefore allowed to neglect this effect - especially if we take the uncertainty in the Gumbel parameters of the observed record into account.

We added the following sentence to the manuscript: "Based on calculations by Rijkswaterstaat (de Goederen, 2013, page 23), and neglecting the effect of the river discharge on Hoek van Holland, the water level at Rotterdam can be approximated by:..."

- *p5. line 27-28: 'River run-off is not considered in this paper'. Do the authors mean it is not considered in the hydrodynamic model? Because later in the paper there is a talk about run-off again.*
  We meant that River run-off is not considered in the hydrodynamic model. We agree that it is misleading, and removed this sentence.

- *fig.6: upper panel – 2 times "observation" in the legend.*
  corrected